# Reaction Mechanism of *CO*_2_ with Choline-Amino Acid Ionic Liquids: A Computational Study

**DOI:** 10.3390/e24111572

**Published:** 2022-10-31

**Authors:** Fabio Ramondo, Simone Di Muzio

**Affiliations:** 1Department of Chemistry, University of Rome ‘La Sapienza’, P.le Aldo Moro 5, 00185 Rome, Italy; 2Istituto dei Sistemi Complessi-Consiglio Nazionale delle Ricerche-ISC-CNR U.O.S. Sapienza, P.le A. Moro 5, 00185 Rome, Italy; 3Department of Physical and Chemical Sciences, University of L’Aquila, Via Vetoio, 67100 L’Aquila, Italy

**Keywords:** CO_2_ capture, ionic liquids, DFT, reaction mechanisms

## Abstract

Carbon capture and sequestration are the major applied techniques for mitigating $CO_{2}$ emission. The marked affinity of carbon dioxide to react with amino groups is well known, and the amine scrubbing process is the most widespread technology. Among various compounds and solutions containing amine groups, in biodegradability and biocompatibility perspectives, amino acid ionic liquids (AAILs) are a very promising class of materials having good $CO_{2}$ absorption capacity. The reaction of amines with $CO_{2}$ follows a multi-step mechanism where the initial pathway is the formation of the C−N bond between the $NH_{2}$ group and $CO_{2}$. The added product has a zwitterionic character and can rearrange to give a carbamic derivative. These steps of the mechanism have been investigated in the present study by quantum mechanical methods by considering three ILs where amino acid anions are coupled with choline cations. Glycinate, L-phenylalanilate and L-prolinate anions have been compared with the aim of examining if different local structural properties of the amine group can affect some fundamental steps of the $CO_{2}$ absorption mechanism. All reaction pathways have been studied by DFT methods considering, first, isolated anions in a vacuum as well as in a liquid continuum environment. Subsequently, the role of specific interactions of the anion with a choline cation has been investigated, analyzing the mechanism of the amine–$CO_{2}$ reaction, including different coupling anion–cation structures. The overall reaction is exothermic for the three anions in all models adopted; however, the presence of the solvent, described by a continuum medium as well as by models, including specific cation- -anion interactions, modifies the values of the reaction energies of each step. In particular, both reaction steps, the addition of $CO_{2}$ to form the zwitterionic complex and its subsequent rearrangement, are affected by the presence of the solvent. The reaction enthalpies for the three systems are indeed found comparable in the models, including solvent effects.

## 1. Introduction

Carbon dioxide, $CO_{2}$, is one of the primary greenhouse gases in the atmosphere. Human activities contribute to an increase in $CO_{2}$ emissions, leading to global warming and climate change [1,2,3]. The development of economic and green technologies for efficient carbon capture and sequestration is a challenge in mitigating $CO_{2}$ emissions [4,5,6]. One of the primary aims of the research is to design materials with high physisorption and chemisorption capacity of carbon dioxide. Chemical reactions of $CO_{2}$ with aqueous amines have long been applied for removing post combustion $CO_{2}$ [7,8,9,10]. However, the high corrosiveness and volatility of these amine solutions are the major disadvantages that are non-ecofriendly and make this technique expensive [11]. Large-scale $CO_{2}$ capture using amines has, therefore, not been widely applied, and effort has been dedicated to the development of materials that could be efficient, environmentally friendly and economic scrubbers of greenhouse gases [12,13]. In the last twenty years, ionic liquids (ILs) have been proposed as alternative solvents for $CO_{2}$ capture [14,15,16,17,18]. Their low vapour pressure, high thermal stability and wide liquid range, along with the ability to tune their physical and chemical properties to specific tasks, provide a practical advantage over amines. The study of specific ionic liquids to be used as scavengers in $CO_{2}$ capture is, therefore, an attractive field of research and, since the study of Blanchard et al. [19] on 1-butyl-3-methylimidazolium hexafluophosphate, a number of potential ILs has been designed to optimise the efficiency of capture [18]. $CO_{2}$ absorption in ILs can involve a simple physical dissolution, physisorption or a reaction between ionic components and $CO_{2}$ chemisorption. Whereas physical absorption becomes relevant only at high pressure, the affinity towards $CO_{2}$ can be enhanced by introducing an amine group to the molecular components of IL that reacts with $CO_{2}$ to form carbamate species [20]. The capacity to form reversible chemical bonds is, therefore, essential to improve the $CO_{2}$ absorption extent of ILs, making them comparable with amines.

Various types of ILs have been studied for $CO_{2}$ capture. Ionic liquids with an acetate anion were some of the first reported $CO_{2}$-reactive ILs [21] by observing that $CO_{2}$ dissolves in the 1-butyl-3-methylimidazolium acetate IL to a great degree. An interesting system with $CO_{2}$ capture efficiency is a class of binary mixtures that transform into an IL upon reaction with CO_2_ (switchable solvents) [22]. Due to their polarity change from a neutral form to a polar IL, their viscosity strongly increases and, therefore, the reaction in these solvents is limited by $CO_{2}$ diffusion. With the aim to reduce the viscosity of ILs, due mainly to intermolecular hydrogen bonding, ILs with aprotic heterocyclic anions, such as N-etherocyclic pyrrolide and pyrazolide, have been proposed and proved suitable for reversible $CO_{2}$ capture [23]. The deprotonation of the imidazolium cation of a conventional imidazolium IL with a superbase, a neutral organic compound that is a stronger Bronsted base than the hydroxide anion, is an alternative means to design $CO_{2}$-capturing ILs [24]

From biodegradability and biocompatibility perspectives, ILs consisting of choline cation (Ch) and anions obtained by deprotonation of amino acids (AA) are particular amine-functionalized ILs [25,26] and have been proposed as efficient $CO_{2}$ capture media [27,28,29,30,31,32,33]. The excellent performance of these AA-based ILs originates from the simultaneous presence in the AA anions of the amino moiety, which allows specific interactions with $CO_{2}$, and the carboxylate group, which can enhance the physisorption of $CO_{2}$. However, the general drawback of AAILs is their high viscosity due to intermolecular hydrogen bonding between ionic components, which further increases owing to the formation of intermolecular hydrogen bonding among AAIL-CO_2_ complexes. The viscosity of ILs is usually reduced by the addition of water, whereas the viscosity increase from reacting with $CO_{2}$ could be avoided by employing species that favour intramolecular over intermolecular hydrogen bonding [34].

Differences in the $CO_{2}$ capture capacity exist between various amine-functionalized materials, and different capture mechanisms may be followed. $CO_{2}$ absorption can occur with 1:0.5 amine/$CO_{2}$ mole ratio (2:1 mechanism), 1:1 amine/$CO_{2}$ mole ratio (1:1 mechanism) or even higher (1:2 mechanism) [27,28,29]. The reaction mechanism of amines with $CO_{2}$ has been studied [28,35,36] and may consist of two steps, as reported in the Figure 1. If the second reaction of Figure 1 is hindered, the absorption of $CO_{2}$ is characterized by a 1:1 stoichiometry with a high absorption capacity; the second step reduces the efficiency since two amine units need to react with only one $CO_{2}$ molecule. Small differences in the local structure of the amine group of AAILs can enhance or hinder the second reaction step and affect the amine/$CO_{2}$ molar fraction.

Previous studies of the reaction mechanism [28,36] showed that the initial step is the formation of a complex between $CO_{2}$ and the amine group with a zwitterionic character. The initial adduct can form via intramolecular proton transfer from $NH_{2}$ to one of the $CO_{2}-$ groups, two alternative carboxylic acids. These compounds can interconvert each other by internal rotation or intramolecular proton transfer to give more stable final products (Figure 1, reaction 1).

Several computational approaches have been applied to examine the reaction mechanism and are summarized in the review by Sheridan et al. [37]. From a detailed study based on ab initio methods, it emerges that the formation of the anion-$CO_{2}$ adduct is the rate-determining step for the carbon dioxide absorption of several AAILs [38]. The small glycinate anion, [Gly], often taken often as a benchmark for modelling larger AA anions, is the most studied system [39,40,41]. The reaction mechanism has been studied for a series of aliphatic AA anions [40,41,42,43] and some multiple amino groups AA anions [44] in a vacuum and in a liquid environment by using the polarizable continuum model (PCM). The inclusion of the cation has been taken into account only in a few examples where the cation tetraalkylphosphonium is coupled with some aprotic heterocyclic [42] or glycinate anions [40].

In this paper, we examine, by quantum mechanical computational methods, the reaction of three AA anions with $CO_{2}$ both in the absence and in the presence of a choline cation to evaluate the effect of specific interactions between cations and anions on the $CO_{2}$ reactivity. Three AA anions have been considered in the present study: the prototype glycinate anion [Gly], an aromatic amino acid anion, phenylalanilate [Phe] and the AA anion having a secondary amino group, prolinate anion [Pro]. Some experimental studies [33,45,46] showed that various [Ch][AA] ILs have different $CO_{2}$ sorption capacities. However, the absorption capacity of pure ionic liquids has not been evaluated due to their high viscosity, further increased upon $CO_{2}$ reaction. Absorption performances have been examined for various [Ch][AA] ILs in dimethyl sulfoxide (DMSO) [33] and aqueous [31,45,46] solutions. For example, [Ch][Gly] IL has the highest sorption with respect to other [Ch][AA] ILs [45] in water solutions; similarly, [Ch][Gly] possesses the highest, whereas [Ch][Phe] has the lowest absorption capacity in DMSO at any concentration [33]. Bearing in mind that the core of $CO_{2}$–amine chemistry is an acid–base reaction, we should take into account that the basicity of the amine is the most critical factor governing the $CO_{2}$-capture performance. However, it is not straightforward to find a direct correlation between the pKa values of the anions and the stability of the AA-IL-CO_2_ complex since pKa mostly describes the basicity of each compound in water, whereas pure IL is a medium with physical and chemical properties deeply different with respect to a water solution. Notwithstanding $CO_{2}$ absorption in ILs being a really complex phenomenon affected by different parameters, among which the solvent and IL concentration, the viscosity change, the surface tension and $CO_{2}$ diffusivity in solution, it is interesting to investigate if different local structural properties of the amine group in amino acid anions can affect some fundamental steps of the absorption mechanism. Within this aim, the reaction (1) of Figure 1 has been studied in detail for the three systems under different conditions. A first step (see the reaction in Figure 2), the interaction of the AA anion with $CO_{2}$, has been examined to evaluate the role of the physisorption and chemisorption components and explore the relative barrier heights. Since this is the rate-determining step [38], it is interesting to investigate if the introduction of an aromatic substituent or the presence of a secondary amine changes the reaction energy with respect to the benchmark reaction of [Gly] with $CO_{2}$. Subsequently, the conversion of the zwitterionic complex in a carbamic acid has been investigated by considering different intramolecular proton transfer pathways and internal rotation processes (see reaction b of Figure 2). All the processes have been studied by DFT methods for the isolated anion in a vacuum and liquid environments (PCM) and for the anion interacting with choline cation within different coupling structures.

## 2. Computational Details

The structure of the glycinate [Gly], L-phenylalanilate [Phe] and L-prolinate [Pro] anions and their ion pairs with the choline cation [Ch] (Figure 1) has been investigated by quantum-mechanical (QM) calculations using the Gaussian 16 package [47].

Molecular geometries and reaction energies were obtained by using density functional theory (DFT) methods. Some points on the potential energy surface (PES) were initially localized employing the B3LYP (Becke’s three-parameter exchange [48] and Lee, Yang and Parr correlation [49] potentials), combined with the double zeta 6-31G* gaussian basis set. The optimised structures were then further investigated through geometry optimizations and frequency calculations by the Minnesota meta hybrid CGA M062X [50] exchange and correlation functional and employing the 6-311++G** basis set. The good reliability of such a function in describing energetic and structural aspects of several liquids has been discussed in earlier reports [51]. Vibrational frequencies were calculated to characterize each critical point and to evaluate thermal corrections (298 K). When needed, intrinsic reaction coordinate (IRC) [52] calculations were carried out to verify that the transition states really connect the corresponding reagents and products.

The liquid environment has been reproduced by repeating the calculation for all the localized structures using the Polarized Continuum Model (PCM) with the parameters of acetonitrile, which has a dielectric constant of 35, slightly higher than that of amino acid-based ionic liquids [53,54].

The reaction of each anion with $CO_{2}$ has been investigated in a vacuum and in liquid phases by considering two steps. In the first step, the thermodynamic stability of the zwitterionic complex with $CO_{2}$ and an energetic barrier to its formation has been determined by fully relaxed PES scansions along the C···N distance. In the second step, the formation of the carbamic derivative has been evaluated by considering an intramolecular proton transfer and its barriers. Reaction energies have been calculated for the isolated anions and for the anions in the presence of the choline cation both in a vacuum and using the PC model.

## 3. Results and Discussion

### 3.1. Choline–Anion Ion Pairs

The association between the choline cation and anion produces the formation of very stable ion pairs, as observed in many cholinium amino acid-based ionic liquids [55] and choline–carboxylate [56,57,58] ionic liquids. Electrostatic attraction between the charged heads of ionic constituents and hydrogen bonding involving the OH group of the choline cation are the main interactions responsible for coupling. In our ionic liquids, among various interaction structures, we show in Figure 1 two stable ion pairs where the OH group coordinates the carboxylate group by OH···O bond (Figure 1a,c,e) or the amino group by OH···N bond (Figure 1b,d,f). For all three anions, we observe an energetic preference for the latter structures where the OH group of choline is hydrogen bonded to the amino group, and the polar heads of the cation and anion are favourably oriented. This solvation process is a factor that limits $CO_{2}$ capture for various reasons. First, it determines the high viscosity of these liquids [59,60] and a consequent low diffusion of $CO_{2}$ within these fluids. However, the strong intermolecular hydrogen bonding observed in dry conditions weakens under humid conditions [61]. Therefore, the addition of water to ILs decreases their viscosity, increases the $CO_{2}$ diffusion in the fluid and enhances its carbon dioxide absorption capacity. Secondly, the efficiency of the reaction with $CO_{2}$ is conditioned by the energy necessary to desolvate the amino group (75 kJ/mol for [Gly][Ch], 88 kJ/mol for [Phe][Ch] and 78 kJ/mol for [Pro][Ch]). However, the association between cations and anions can also occur through the alternative structures in Figure 1a,c,e that have quite a similar coupling energy. They show multiple coordination between the carboxylate group, and the cation without the involvement of the amino group that could, therefore, interact and react with $CO_{2}$. Solvation in these fluids predicts the various association structures of comparable stability; some of them show anions having amino groups not strongly solvated by cations and, therefore, able to be attacked by $CO_{2}$.

### 3.2. Reaction of Anions with $CO_{2}$

As well known [28], the reaction with $CO_{2}$ involves the amino group and it has, as an initial step, the production of a zwitterionic complex through the formation of an N−C bond. $CO_{2}$ addition to the anion has been investigated here through a series of geometry optimizations carried out by approaching a $CO_{2}$ molecule to the $NH_{2}$ group at C···N distances that are progressively shorter. The potential energy profiles obtained for [Gly] (Figure 2a,b), [Phe] (Appendix A) and [Pro] (Figure 3a,b) anions are compared with the analogue curves for [Gly][Ch] (Figure 2c,d), [Phe][Ch] (Appendix A) and [Pro][Ch] (Figure 3c,d) ion pairs. There are a number of considerations emerging from such calculations. The interaction with $CO_{2}$ occurs initially by van der Waals intermolecular interactions, where $CO_{2}$ keeps its linearity, and it is closely physisorbed and not covalently bonded to the anion. The progressive decrease in c the C···N distance gives the formation of the carbamate with a C−N bond distance of about 1.6 Å. The energy of the reaction of the carbamate compound has been calculated with reference to the separate $CO_{2}$ and anions, and their values are reported in Figure 2, Figure 3 and Appendix A. In a vacuum, the reaction of the anion with $CO_{2}$ is exothermic for all three anions without appreciable energetic barriers, and their values are quite similar for [Phe] (66 kJ/mol) and [Pro] (67 kJ/mol) and slightly lower for [Gly] (54 kJ/mol). The reaction is again exothermic for the anions in a liquid environment (PCM) but with substantial differences with respect to the vacuum calculations. [Gly] and [Phe] show a less exothermic reaction (26 and 39 kJ/mol, respectively), and the passage from physisorbed to chemisorbed $CO_{2}$ shows the appearance of small energetic barriers (see Figure 2b and Appendix A); the potential energy profile of [Pro] is again affected by the environment, but the energetic barrier is negligible, as found in a vacuum (Figure 3b). We have reproduced the structure of the saddle points relative to these barriers for each anion (Figure 2, Figure 3 and Appendix A) where the initial formation of the CN carbamate bond and the loss of linearity of $CO_{2}$ is evident. As expected, the main effect caused by the presence of the solvent on this bimolecular reaction is a greater stabilization of the reactants with respect to the condensed final product because of the presence of the two solvation cages.

The formation of adducts with $CO_{2}$ has also been simulated in the presence of the choline cation, considering the ion pair in a vacuum as well as in a liquid environment with a series of PES scansions, as in the case of the anion. Starting from the structure where the cation and anion are coupled by an OH···O hydrogen bond, we have analysed the energy of the system by progressively making $CO_{2}$ approach the $NH_{2}$ group. The presence of the cation deeply changes the energy of the reaction of [Gly][Ch] and [Phe][Ch] with $CO_{2}$: in these cases, the reaction is slightly endothermic when considered in a vacuum (Figure 2c and Appendix A). It means that the formation of a carbamate bond is not the favourite when the cation and anion are strongly coupled; the interaction with $CO_{2}$ gives an adduct in which $CO_{2}$ is weakly bonded to the anion. This result is in line with the experimental evidence that $CO_{2}$ absorption is favoured by the dilution of AAILs in dimethyl sulfoxide (DMSO) [33]. Strong ionic coupling in ILs hinders the reaction with $CO_{2}$, whereas the addition of a solvent causes an increase in the distances in the ionic couples and can affect the absorption mechanism. The reaction is again found to be exothermic when $CO_{2}$ and the ion pair interact in the liquid environment, although the carbamate formation energy is clearly less favourable than that observed for the anions alone (Figure 2d and Appendix A), and small activation barriers have been calculated. The saddle points relative to each reaction are reported in Figure 2, Figure 3 and Appendix A. In particular, the reaction of [Gly] with $CO_{2}$ in the presence of the cation seems to be less favourable (7 kJ/mol) than that of [Phe] (28 kJ/mol). $CO_{2}$ instead continues to be significantly chemisorbed to the prolinate anion, giving exothermic reactions with energies very similar in a vacuum and in a liquid (Figure 3c,d). It means that the capture of $CO_{2}$ from the prolinate anion is energetically favourable and not drastically hindered by specific interactions with cations.

### 3.3. Formation of Carbamic Derivatives for the Glycinate and L-Phenylalanilate Anions

The initial zwitterionic complex can evolve towards the formation of a carbamic derivative through intramolecular pathways [28]. This mechanism and its energy have been considered for the three anions studied here, first in the absence of the choline cation. The zwitterionic complex will be indicated from now on as $\mathrm{Gly}CO_{2}$ for glycinate, $\mathrm{Phe}CO_{2}$ for phenylalanilate and $\mathrm{Pro}CO_{2}$ for prolinate anions. For [Gly] and [Phe], we have considered two possible reaction pathways involving intramolecular proton transfer from the amino group towards the carboxylate groups. The products formed are different depending on which carboxylate accepts the proton; the reaction pathways and their energies for [Gly] are reported in Figure 4. In one pathway (path 1 in Figure 4A), the proton moves onto the new ${CO}_{2}^{-}$ group added by the reaction with $CO_{2}$ and makes carbamic acid, here indicated by $\mathrm{Gly}CO_{2}H$(1A). The reaction is exothermic, but the proton transfer barrier height is quite high, with the transition state (TS) having a cyclic structure with four atoms.

The alternative pathway (path 2 in Figure 4A) predicts the proton transfer onto the other carboxylate group to form $\mathrm{Gly}CO_{2}H$(2A) through a five-membered transition state (TS1); the reaction is again exothermic, but it shows very low activation energy. The two products, $\mathrm{Gly}CO_{2}H$(1A) and $\mathrm{Gly}CO_{2}H$(2A), have comparable stability, but the different accessibility of the transition states gives the second pathway as the only accepted mechanism to capture $CO_{2}$. The $\mathrm{Gly}CO_{2}H$(2A) is followed by a first internal rotational about the C−C bond that favours the formation of an intramolecular hydrogen bond between carboxylic and carboxylate groups $\mathrm{Gly}CO_{2}H$(2B) and gives a strong stabilization of the product. The seven-membered cyclic structure allows fast proton transfer and the formation of the product $\mathrm{Gly}CO_{2}H$(1B); that is instead hindered through the first mechanism, the direct proton transfer from $NH_{2}$. Internal rotation and proton transfer lead, therefore, to the formation of two isomers that have nearly equivalent energies. The introduction of the solvent as a continuum does not change the energetic profile of both mechanisms calculated for the isolated anion, as summarized in Figure 4B. In addition, we have investigated the flexibility of the zwitterionic compound by considering that rotation about the CN bond of the carbamate group could bring the interconversion of $\mathrm{Gly}CO_{2}H$(1A) to $\mathrm{Gly}CO_{2}H$(1B). The energetic profile, calculated through a fully relaxed PES scan in the liquid environment, is reproduced in Appendix A, along with all the stationary points localized during the scan. It is plain that torsions about the CN bond are strongly hindered since its partial double-bond character: it means that even if the reaction followed the first pathway, the highest barrier one, product $\mathrm{Gly}CO_{2}H$(1A) would have no way to easily interconvert to the most stable $\mathrm{Gly}CO_{2}H$(1B). The second pathway seems, therefore, to indeed be the only mechanism to form a carboxylic group from the initial zwitterionic complex.

These reaction pathways have been calculated for the second anion, phenylalanilate, both isolated and in a solvent medium, as summarized in Figure 4. The structural features of the reagents, products and transition states, shown in Figure 5A, localized for this system are indeed similar to those of the glycinate anion with an energetic profile comparable, see Figure 5B. The only difference is the lack of a barrier to the intramolecular proton transfer between two ${CO}_{2}^{-}$ groups ($\mathrm{Phe}CO_{2}H$(2B) and $\mathrm{Phe}CO_{2}H$(1B)).

The influence of the choline cation on the reaction mechanism has been evaluated by the following calculations. The zwitterionic complex formed by the addition of $CO_{2}$ can be solvated through different interaction structures. Figure 6 shows two models, [$\mathrm{Gly}CO_{2}$][Ch](A), the most stable one, and [$\mathrm{Gly}CO_{2}$][Ch](B), where choline is hydrogennbonded alternatively to one of the ${CO}_{2}^{-}$ groups. The first structure, [$\mathrm{Gly}CO_{2}$][Ch](A), is suitable for evaluating the role of explicit interactions with the cation on the energy of the first mechanism. The reaction continues to be weakly exothermic with a high barrier to proton transfer, as predicted for the isolated anion. Product [$\mathrm{Gly}CO_{2}H$][Ch] can then undergo a structural arrangement by internal rotation to make an intramolecular hydrogen bond that stabilizes the final product. The second structure, [$\mathrm{Gly}CO_{2}$][Ch](B), is instead a good starting point to investigate the second mechanism in the presence of the cation. This reaction pathway again has a transition state that is more accessible and, in addition, is more exothermic with respect to the reaction in the absence of choline. Once again, internal rotations about C−N bonds can give orientations favourable to the formation of the intramolecular hydrogen bond. The fast intramolecular proton transfer between carboxylic and carboxylate groups predicted for the isolated anion is instead hindered by the specific interactions with choline. Once transferred from the $NH_{2}$ group, the proton remains well localized in one group without being easily transferred between the oxygen atoms.

The same considerations can be derived from the examination of the reaction pathways for the [$\mathrm{Phe}CO_{2}$][Ch] ion pair, as shown in Figure 7. As for the glycinate anion, the mechanism requiring a strained four-membered ring transition state can be excluded, and the product [$\mathrm{Phe}CO_{2}H$][Ch] is obtained through a barrier height (15 kJ/mol) comparable with that calculated in the absence of choline cation. Structural rearrangements through rotation about the C−N bond give further stabilization of the product by an intramolecular hydrogen bond.

### 3.4. Formation of Carbamic Derivatives for the L-prolinate Anions

On the basis of the results obtained for [Gly] and [Phe] anions, we have considered only the more favourable reaction pathway (path 2), which involves intramolecular proton transfer to the carboxylate group of prolinate, see Figure 8A.

The energy barrier (Figure 8B) is comparable with those calculated for glycinate and L-phenylalanilate anions, whereas product $\mathrm{Pro}CO_{2}H$(2A) has been found to be only 6 kJ/mol below with respect to the starting adduct $\mathrm{Pro}CO_{2}$. The reaction seems, therefore, to be less exothermic than the corresponding reactions predicted for [Gly] and [Phe]. However, the easy rotation of the C-N bond of prolinate (very low energy barrier) allows a strong stabilization of the products by intramolecular hydrogen bonding $\mathrm{Pro}CO_{2}H$(2B). The alternative product, $\mathrm{Pro}CO_{2}H$(1B), where protonation occurs at the other ${CO}_{2}^{-}$ group, has once again comparable stability and can be obtained from $\mathrm{Pro}CO_{2}H$(2B) by intramolecular proton transfer with a small barrier height. This process is instead found to be barrierless in the solvent phase. In addition, we can observe that the incorporation of the continuum solvation model makes the overall reaction less exothermic, as found for the glycine anion, and it could be due to intramolecular hydrogen bonding that produces a charge delocalization with a consequent decrease in the zwitterionic character with respect to the starting $\mathrm{Pro}CO_{2}$ compound and less stabilization in the solvent phase.

This mechanism of proton transfer from $NH_{2}$ to ${CO}_{2}^{-}$ of the proline anion is then considered again in the presence of choline and, including the continuum solvation model by starting from two ion pairs, each representative of the interactions of the cation anion through hydrogen bonding (see Figure 9). In [$\mathrm{Pro}CO_{2}$][Ch](A), more stable than [$\mathrm{Pro}CO_{2}$][Ch](B) by about 10 kJ/mol, the OH···O bond involves the ${CO}_{2}^{-}$ group of prolinate. Intramolecular proton transfer from $NH_{2}$ shows a barrier energy higher than in the absence of the cation and gives a product nearly isoenergetic with the starting compound. However, a simple rotation of the $CO_{2}H$ allows a fast intramolecular proton transfer from ${CO}_{2}^{-}$ of prolinate to the second ${CO}_{2}^{-}$ to give the final product found 21 kJ/mol lower than initial ion pair. The relative transition state, reproduced in Figure 9, has been found in a vacuum but not in a solvent phase, where the proton is transferred without a barrier height. The alternative pathway starting from [$\mathrm{Pro}CO_{2}$][Ch](B) describes the same mechanism of intramolecular proton transfer when choline interacts with $CO_{2}$ from prolinate. The barrier height is comparable with the previous one, but this reaction is exothermic (15 kJ/mol). The product can easily orient the ${CO}_{2}^{-}$ and $CO_{2}H$ groups to produce a strong intramolecular hydrogen bond, and the final product was found to be 31 kJ/mol below the initial zwitterionic adduct. We can, therefore, conclude that proton transfer from $NH_{2}$ to ${CO}_{2}^{-}$ is an exothermic reaction with a modest barrier height. In the absence of the choline cation, the proton can easily jump from one ${CO}_{2}^{-}$ to another, whereas the specific interaction of choline with ${CO}_{2}^{-}$ hinders this proton transfer. In conclusion, the overall reaction of [Pro] with $CO_{2}$ is found to be exothermic, with values ranging from 70 to 80 kJ/mol: the initial formation of zwitterionic complex gives a marked stabilization followed by further energy gain for the rearrangement of the addition product to produce carboxylic acid. The presence of the choline cation has a slight effect on the reaction of the formation of zwitterionic adduct, whereas it plays a more relevant role in the subsequent proton transfer reactions.

## 4. Conclusions

The mechanism of the reaction of three amino acid anions with $CO_{2}$ has been studied by quantum mechanical methods under different conditions. The glycinate, L-phenilalanilate and L-prolinate anions have been considered with the aim of evaluating if differences in the local structure of the amine group of the AA anion can affect the energy of their reaction with $CO_{2}$. We initially examined the interaction of $CO_{2}$ with the AA anion to evaluate the role of the physisorption and chemisorption components by exploring the potential energy surface at different C···N distances between $NH_{2}$ and $CO_{2}$ units. Subsequently, the addition product between AA anion and $CO_{2}$ can undergo various intramolecular proton transfers from $NH_{2}$ to one of the $CO_{2}$ groups and give the formation of different carbamic derivatives. The energy of each reaction has been calculated for the isolated anions and for the anions in the presence of a surrounding continuum medium and repeated for the anions in the presence of the choline cation. The overall reaction is exothermic for the three anions considered in all calculation models, however, the presence of the solvent can modify the value of the reaction energy, as summarized in Table 1. The first step, the formation of the zwitterionic addition product, is an exothermic barrierless process in a vacuum with energy comparable for the three anions. The presence of the solvent, described by continuum models, lowers the exothermicity of the reaction and suggests a higher reactivity of [Pro] with respect to [Gly] and [Phe]. The second step that gives the formation of the carbamic acid is still an exothermic process that, in the liquid continuum environment, shows energy comparable for the three systems. By considering the contribution of both steps to the overall reaction, we can observe that the [Gly], [Phe] and [Pro] can react with $CO_{2}$ by exothermic processes and give the formation of carbamic acid. Within the polarized continuum description of the IL, we observe a higher affinity of [Pro] toward $CO_{2}$. However, the introduction of the choline cation and the specific interactions with the anion change the energy of both steps of the reaction mechanism. In particular, the intramolecular proton transfer described in the second step and the subsequent rearrangement of the carbamic acid is affected by the presence of the cation. In conclusion, the role of the cation is not negligible when the overall mechanism is considered, and the value of the enthalpy of the reaction is found to indeed be comparable for the three systems. In addition, it is reasonable to expect that the intramolecular hydrogen bond occurring in the reaction products and involving the carboxylic group should reduce its acidity and, therefore, hinders reaction (2) of Figure 1.

## Data Availability

Not applicable.

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
