# Peer review of "Reaction Mechanism of CO2 with Choline-Amino Acid Ionic Liquids: A Computational Study"

_entropy, 2022, doi:10.3390/e24111572_

Round 1
Reviewer 1 Report
1. Is there any effect of anion for the stability of AA-IL-CO2 complex formation and its stability ?
2. Explain the stabilization factor for AA-IL-CO2 complex formation.
3. How to you explain the catalytic role of AA-based ILs for AA-CO2 complex formation ?
4. Is there any effect of structural modification (L or D conformation ) of AA based ILs on AA-CO2 complex formation ?
5. Is there any role of hydroxyl group of choline on stability of AA-CO2 complex formation ?
Author Response
Referee 1
- Is there any effect of anion for the stability of AA-IL-CO2 complex formation and its stability ?
On the basis of our calculations, we can conclude that the role of the anion on the stability of AA-IL-CO2 complex does not easily emerge and this is already summarized in the conclusions section. In absence of specific interactions with the cation, we found that the prolinate anion shows the higher affinity to CO2 and the overall reaction is more exothermic. The presence of the cation has instead an important role on the second step, the rearrangement of the zwitterionic complex, and, by considering the effect of both the steps, we find that three systems shows reaction enthalpies comparable.
- Explain the stabilization factor for AA-IL-CO2 complex formation.
The following consideration has been added in the introduction section as follows
“Bearing in mind that the core of CO2–amine chemistry is an acid–base reaction, we should take into account that the basicity of amine is the most critical factor governing the CO2-capture performance. However, it is not straightforward find a direct correlation between the pKa values of the anions and the stability of the AA-IL-CO2 complex, since mostly pKa describes the basicity of each compound in water whereas pure IL is a medium with physical and chemical properties deeply different with respect to a water solution.”
- How to you explain the catalytic role of AA-based ILs for AA-CO2 complex formation ?
The reactivity of AA-based ILs towards CO2 can be found in the fact that amino acids in ILs are present as anionic form. It is interesting to note that amino acids simply dissolved in water show almost no CO2 absorption, since their amino groups are usually protonated in water. However, after deprotonating -NH3 + with alkaline substances (e.g., KOH), AAS solvents can react with CO2 in a manner similar to alkanolamines, [B. Jiang, X. Wang, M.L. Gray, Y. Duan, D. Luebke, B. Li, Development of amino acid and amino acid-complex based solid sorbents for CO2 capture, Applied Energy 109 (2013) 112–118].
- Is there any effect of structural modification (L or D conformation) of AA based ILs on AA-CO2 complex formation ?
Additional calculations were carried out on the reaction of D-phenylalanilate and D-prolinate anions with CO2 and the subsequent rearrangement of the zwitterionic adduct but the energetic patterns is indeed similar with those of the L-isomers here discussed. The results of the calculations are not reported in the manuscript.
- Is there any role of hydroxyl group of choline on stability of AA-CO2 complex formation ?
From our liquid environment (PCM) calculations, it emerges that choline interacts through OH…O hydrogen bonding mainly with carboxylate group and this interaction does not drastically change the exothermicity of the CO2 addition reaction. We should take into account that also the OH group is potentially an active site able to react with CO2 but previous studies [S. Kim et al. / International Journal of Greenhouse Gas Control 45 (2016) 181–188] ∗on the reaction mechanism of CO2 with ethanol-amines showed that the hydroxyl part was less efficient as a reactant or as a catalyst than the amino group.

Reviewer 2 Report
In this work of Ramondo et. al., detailed theoretical study on the CO2 chemisorption behavior of AAILs has been conducted. Minor revision is recommended:
1. In the introduction part, not only the advantages of the amino-derived ILs in CO2 sorption, but also the unsolved issues should be mentioned to give a full understanding in related fields.
2. Indeed, extesnive studies have been done on developing ILs towards efficient carbon capture. Particularly, in terms of ILs, there are other types besides amino-based ILs, which should be included.
3. In the abstract, the author only pointed out the IL systems they worked on, but had no conclusion on what has been obtained through their calculation and what guidance they have provided for future ILs design.
4. As [Ch] contained a hydroxyl group and may influence the deprotonation step, this should be illustrated.
Author Response
Referee 2
In this work of Ramondo et. al., detailed theoretical study on the CO2 chemisorption behavior of AAILs has been conducted. Minor revision is recommended:
- In the introduction part, not only the advantages of the amino-derived ILs in CO2 sorption, but also the unsolved issues should be mentioned to give a full understanding in related fields.
The general drawback of most ILs and in particular of AAILs is their high viscosity that further increases upon CO2 interaction. This point has been already mentioned in the 3.1. Now is reported also in the Introduction along with a mention to some example of AAILs where the increase of viscosity is reduced [34]. This is the added text in the introduction:
“However, the general drawback of AAILs is their high viscosity, due to intermolecular hydrogen bonding between ionic components, that further increases owing to formation of intermolecular hydrogen bonding among AAIL-CO2 complexes. The viscosity of ILs is reduced by addition of water whereas the viscosity increase from reaction with CO2 could be avoided employing species that favour intramolecular over intermolecular hydrogen bonding [Ref. 34]”
- Indeed, extesnive studies have been done on developing ILs towards efficient carbon capture. Particularly, in terms of ILs, there are other types besides amino-based ILs, which should be included.
As suggested by the Referee we now mention different ILs proposed as CO2 capture solvents with the following sentences added to the previous version of the Introduction section.
“Various types of ILs have been studied to CO2 capture. Ionic liquids with the acetate anion were some of the first reported CO2 reactive ILs [21] by observing that CO2 dissolves in the 1-butyl-3-methylimidazolium acetate IL to a great degree. An interesting system with CO2 capture efficiency is a class of binary mixtures that transform into a IL upon reaction with CO2 (switchable solvents) [22]. Due to their polarity change, from a neutral form to a polar IL, their viscosity strong increases and, therefore, reaction in these solvents is limited by CO2 diffusion. With the aim to reduce viscosity of ILs, due mainly to intermolecular hydrogen bonding, ILs with aprotic heterocyclic anions, like N-etherocyclic pyrrolide and pyrazolide, has been proposed and they proved suitable for reversible CO2 capture [23]. Deprotonation of the imidazolium cation of a conventional imidazolium IL with a superbase, a neutral organic compound that is a stronger Bronsted base than the hydroxide anion, is an alternative mean to design CO2 capture ILs [24]”
- In the abstract, the author only pointed out the IL systems they worked on, but had no conclusion on what has been obtained through their calculation and what guidance they have provided for future ILs design.
Some considerations about the results of the study have been added in the last part of the Abstract.
“The overall reaction is exothermic for the three anions in all the models adopted, however the presence of the solvent, described by a continuum medium as well as by models including specific cation- anion interactions, modifies the values of the reaction energies of each step. In particular, both the reaction steps, the addition of CO2 to form the zwitterionic complex and its subsequent rearrangement, are affected by the presence of the solvent. The reaction enthalpies for the three systems are found indeed comparable in the models including solvent effects”
- As [Ch] contained a hydroxyl group and may influence the deprotonation step, this should be illustrated.
This comment suggests us to discuss if the OH group of the Choline may be involved in the whole reaction with CO2. Our calculations show that choline cation strongly interacts with the zwitterionic complex mainly involving the carboxylate group. This interaction seems to play a relevant role only upon intramolecular proton transfer from NH2 group to CO2- groups to form carboxylic group. The rearrangement of the reaction product is driven by the formation of Intramolecular hydrogen bonding between carboxylic and carboxylate groups where the quick proton exchange, expected in the gas phase, is prevented in the presence of the cation. As concerning the reaction (2) of the Scheme 1, the deprotonation of the carboxylic group, it is reasonable to expect that intramolecular hydrogen bonding involving the carboxylic group should reduce its acidity and therefore hinders this reaction step. A short consideration regarding this point is now present in the last part of the conclusions.